# Learning Effective Language Representations for Sequential Recommendation via Joint Embedding Predictive Architecture

**Minh-Anh Nguyen, Dung D. Le**
College of Engineering and Computer Science, VinUniversity, Vietnam
{minh.na2, dung.ld}@vinuni.edu.vn

## Abstract

Language representation learning has emerged as a promising approach for sequential recommendation, thanks to its ability to learn generalizable representations. However, despite its advantages, this approach still struggles with data sparsity and a limited understanding of common-sense user preferences. To address these limitations, we propose **JEPA4Rec**, a framework that combines **J**oint **E**mbedding **P**redictive **A**rchitecture with language modeling of item textual descriptions. JEPA4Rec captures semantically rich and transferable representations, improving recommendation performance and reducing reliance on large-scale pre-training data. Specifically, JEPA4Rec represents items as text sentences by flattening descriptive information such as *title, category*, and other attributes. To encode these sentences, we utilize a bidirectional Transformer encoder with modified embedding layers specifically designed to capture item information in recommendation datasets. We apply masking to text sentences and use them to predict the representations of the unmasked sentences, helping the model learn generalizable item embeddings. To further enhance recommendation performance and language understanding, we employ a two-stage training strategy that incorporates self-supervised learning losses. Experiments on six real-world datasets demonstrate that JEPA4Rec consistently outperforms state-of-the-art methods, particularly in cross-domain, cross-platform, and low-resource scenarios.

## 1 Introduction

Sequential recommendation predicts the next item a user is likely to interact with based on their past behavior. Traditional ID-based methods capture sequential patterns well but struggle with cold-start items and knowledge transfer across domains (Fang et al., 2020; Kang & McAuley, 2018; Sun et al., 2019). To address this issue, cross-domain methods leverage overlapping users or items (Tang et al., 2012; Zhu et al., 2021b), however, their real-world applicability is limited due to the scarcity of shared data. Another approach utilizes modalities such as text or images, but the semantic gap between domains remains a challenge (Yuan et al., 2023). Pretrained language models (PLMs), trained on general data like Wikipedia (Devlin et al., 2019), often fail to align with item descriptions and generate embeddings at the sentence level, limiting their ability to effectively model user preferences (Liu et al., 2023).

Our goal is to leverage language representation learning for sequential recommendation while utilizing PLM knowledge. This involves three key challenges: (1) Creating a flexible item text representation beyond simple attribute concatenation (Ding et al., 2021; Wang et al., 2024). (2) Learning both item sequences and common-sense user preferences for better generalization. (3) Designing an efficient training strategy to bridge the text-recommendation semantic gap (Li et al., 2023b) while maintaining effectiveness in sparse data settings.

We leverage Joint Embedding Predictive Architecture (JEPA) (Assran et al., 2023; Abdelfattah & Alahi, 2024) to address these challenges. JEPA predicts abstract representations rather

than raw tokens, capturing meaningful semantics while avoiding low-level noise (LeCun, 2022). It consists of an encoder and a predictor: the encoder generates latent representations from context-target pairs, while the predictor learns to map context to target representations. Unlike contrastive learning (Jaiswal et al., 2020), JEPA eliminates negative samples and prevents representation collapse through an asymmetric encoder design (Chen & He, 2021). Despite its success in vision tasks, JEPA remains unexplored in Natural Language Processing and recommendation (Gui et al., 2024).

Building upon this, we propose **JEPA4Rec**, a framework that leverages JEPA for language representation learning in sequential recommendation. To enrich the semantic meaning of item representations, we transform item metadata (e.g., title, category, and description) into a single text sentence (illustration in Figure 1). Effectively learning item sentence representations and capturing common sense in user preferences is

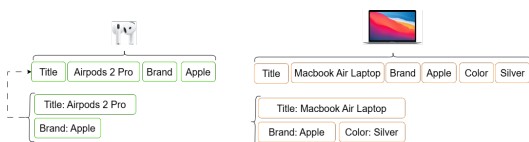

Figure 1: We flatten metadata attributes and corresponding values to represent items

crucial for recommendations. Here, common sense refers to the model's ability to grasp inherent user behaviors, such as brand loyalty and category preferences, beyond explicit interaction patterns. This is where JEPA plays a central role, enabling structured and transferable representation learning. Our main contributions are as follows:

1. We construct a bidirectional Transformer encoder with modified embedding layers tailored for encoding item sentences, which are then used as both the Context and Target Encoders. Additionally, we employ a tokens masking strategy that selectively hides history item information at varying rates. This requires the model to reconstruct missing details from partial item information, allowing generalizable item representations and enhancing common-sense preference learning.

2. We adopt a two-stage training approach (pre-training and fine-tuning), leveraging self-supervised learning objectives tailored for both recommendation and language understanding tasks.

3. Extensive experiments on real-world datasets demonstrate that JEPA4Rec consistently outperforms state-of-the-art methods, achieving significant improvements in recommendation performance across all datasets. Notably, JEPA4Rec requires only a fraction of the pre-training data typically used in previous studies, showcasing its efficiency in learning transferable, robust, and data-efficient item representations, particularly in cross-domain, cross-platform, and low-resource settings.

## 2 Related work

**Sequential Recommendation** aims to predict users' next interactions by modeling historical behaviors. Traditional methods, including RNNs (Hidasi et al., 2015; Li et al., 2017), CNNs (Tang & Wang, 2018), and Transformers (Sun et al., 2019; Assran et al., 2023), rely on item IDs, limiting transferability. A key limitation of this research direction is its reliance on discrete item IDs, which are inherently non-transferable across domains and incapable of capturing the rich semantic information embedded in item metadata. In contrast, textual item representations offer a more generalizable and semantically meaningful alternative. Approaches like (Hou et al., 2022; Geng et al., 2022) use PLMs to embed item descriptions, allowing knowledge to transfer across platforms and domains. However, these methods often separate representation learning from user modeling, limiting their flexibility. JEPA4Rec addresses this gap by jointly learning from item metadata in natural language form, enabling rich semantic understanding and transfer learning without reliance on shared item IDs or overlapping users (Zhu et al., 2021a; Tang et al., 2012). This makes it particularly suitable for cold-start or low-resource scenarios.

**Transfer Learning for Recommendation** addresses data sparsity by leveraging shared knowledge (Singh, 2020). PLMs generate universal item representations (Devlin et al., 2019; Geng et al., 2022) but require large-scale pre-training (Liu et al., 2023) and tightly couple text

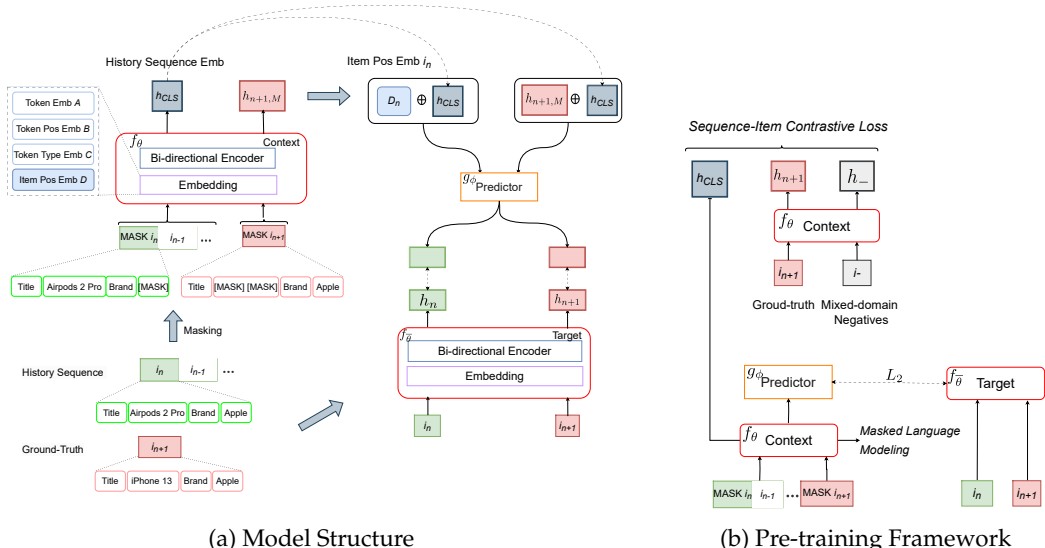

(a) Model Structure       (b) Pre-training Framework

Figure 2: Overview of JEPA4Rec. **Left:** The items are represented as text sentences. The model leverages the user's historical interactions along with partial text information to predict the full representations of the items. Specifically, the historical item sequence embedding $\mathbf{h}_{CLS}$ is combined with the item position embedding $\mathbf{D}_n$ of item $i_n$ and the embedding of the next item $\mathbf{h}_{n+1,M}$ when they are masked. These representations are then passed through the predictor to reconstruct the complete representations of $i_n$ and $i_{n+1}$, which are obtained from the Target Encoder. **Right:** JEPA4Rec includes the Context Encoder ($f_\theta$), which encodes masked item sequences; the Target Encoder ($f_{\bar{\theta}}$), which processes the full item information and is updated via Exponential Moving Average (EMA) of the Context Encoder; and the Predictor ($g_\phi$), which aims to recover the complete item representations from the partially observed inputs.

and item representations (Hou et al., 2023; 2022). Unlike contrastive loss, which encourages instance discrimination, or MLM, which focuses on local token-level semantics, JEPA4Rec introduces a novel L2 Mapping Loss that directly aligns user behavior embeddings with item embeddings. This predictive alignment enables the model to reason in the embedding space rather than over discrete tokens, allowing it to capture user preferences more effectively. Through this architecture, JEPA4Rec can reconstruct complete item semantics from partial textual cues, such as fragments of product descriptions or missing metadata. Furthermore, the model can learn representations that generalize across different domains, even when pre-training data is limited. This embedding-level learning paradigm facilitates better user intent modeling, improves robustness to sparse input, and enhances cross-domain performance without requiring overlapping users or shared item identifiers.

## 3 Methodology

In this section, we introduce JEPA4Rec, a framework designed for efficient language representation learning in sequential recommendation. Building upon Li et al. (2023a), which employs a bidirectional Transformer encoder for encoding item sentences, JEPA4Rec enhances common-sense learning in user preferences and improves generalizable item embeddings through the Joint Embedding Predictive Architecture (JEPA). As illustrated in Figure 2, JEPA4Rec incorporates two key innovations. First, its **masking strategy** transforms user history into text sequences and applies Masked Language Modeling (MLM). However, instead of solely learning at the token level, JEPA4Rec learns item representations in the embedding space, as detailed in Section 3.3.1. Second, its **learning framework** trains a Predictor to reconstruct full item representations using history embeddings and masked item embeddings. The Target Encoder refines these representations, while self-supervised

losses further enhance recommendation accuracy and language understanding, as discussed in Section 3.3.2.

## 3.1 Problem Formulation

We address sequential recommendation with multi-domain interaction data for training or pre-training. A user's history in each domain forms a sequence $s = \{i_1, i_2, \ldots, i_n\}$, where each item $i$ has a unique ID and textual attributes (title, category, description). To preserve domain-specific semantics Hou et al. (2022), we keep sequences separate rather than merging them. Instead of item IDs, we represent items using textual metadata by flattening attributes $a$ (e.g., Title, Brand) and their values $v$ (e.g., iPhone, Apple). Each item is expressed as a sentence $S_i = \{a_1^i, v_1^i, \ldots, a_m^i, v_m^i\}$, where $m$ varies based on available metadata (Figure 1). The model then learns user preferences from the sentence sequence $s = \{S_1, \ldots, S_n\}$ and predicts the next item sentence $S_{n+1}$.

## 3.2 Item representation

**Model Inputs** After flattening the text metadata attributes of an item, we obtain its corresponding sentence representation $S_i$. In sequential recommendation, the most recently interacted items carry the most relevant information about a user's latest preferences (Liu et al., 2021). Consequently, we represent the interaction history as:

$$X = \{[CLS], S_n, .., S_1\}$$

Where $[CLS]$ is a special token used to generalize the sequence information. $X$ is then fed into the encoders.

**Encode Item Mechanism** To encode item sentences, we follow previous works (Sun et al., 2019; Geng et al., 2022) and construct four types of embeddings. First, **token embeddings** $\mathbf{A} \in \mathbb{R}^{V_w \times d}$ represent the meaning of each word token, where $V_w$ is the vocabulary size and $d$ is the embedding dimension. Unlike ID-based approaches (Hua et al., 2023), JEPA4Rec uses text tokens, making the representation more flexible and domain-independent. Second, **token position embeddings** $\mathbf{B} \in \mathbb{R}^d$ capture the position of each token in a sentence, helping the model learn word order. Third, **token type embeddings** $\mathbf{C}_{CLS}, \mathbf{C}_{\text{Attribute}}, \mathbf{C}_{\text{Value}} \in \mathbb{R}^d$ indicate whether a token belongs to the [CLS] token, an attribute key, or an attribute value, which is crucial for handling repeated structures in item metadata. Finally, **item position embeddings** $\mathbf{D} \in \mathbb{R}^{n \times d}$ encode the position of items in a user's interaction sequence, where each item $S_i$ shares the same vector $\mathbf{D}_i$, aligning all tokens within an item and facilitating sequence modeling. Details about the tokenizer can be found in Appendix B.

The input embedding for each word $w$ in sequence $X$ is obtained by summing four embeddings and applying layer normalization (Ba et al., 2016):

$$\mathbf{E}_w = \text{LayerNorm}\left(\mathbf{A}_w + \mathbf{B}_w + \mathbf{C}_w + \mathbf{D}_w\right)$$

The final input representation $\mathbf{E}_X$ consists of these embeddings for all tokens in $X$, including the special $[CLS]$ token:

$$\mathbf{E}_X = \left[\mathbf{E}_{[CLS]}, \mathbf{E}_{w_1}, \ldots, \mathbf{E}_{w_l}\right]$$

Where $l$ is the maximum sequence length. To encode $\mathbf{E}_X$, we use Longformer (Beltagy et al., 2020), a bidirectional Transformer optimized for long sequences. Similar to Longformer's document processing setup, the special token $[CLS]$ has global attention, while other tokens rely on local windowed attention. The model generates $d$-dimensional word representations as follows:

$$\left[\mathbf{h}_{CLS}, \mathbf{h}_{w_1}, \ldots, \mathbf{h}_{w_l}\right] = \text{Encoder}\left([\mathbf{E}_{[CLS]}], \mathbf{E}_{w_1}, \ldots, \mathbf{E}_{w_l}\right)$$

where each word representation $\mathbf{h}_w \in \mathbb{R}^d$. Following standard language model practices, we use $\mathbf{h}_{CLS}$ as the sequence representation. To encode items, JEPA4Rec treats each item as a single-item sequence $X = \{[CLS], S_i\}$ and obtains its embedding $\mathbf{h}_i$ from the sequence representation. We use the same embedding layer and Longformer encoder for both the

Context Encoder $f_\theta$ and Target Encoder $f_{\bar\theta}$, with the latter updated via an exponential moving average for stable training and to prevent representation collapse (Chen & He, 2021).

### 3.3 Pre-training Framework

The goal of pre-training is to establish a strong parameter initialization for downstream tasks. Our approach integrates both language understanding and recommendation learning and enhances the model's ability to learn common sense in user preferences, improve item representation, and create generalizable item embeddings without relying on large-scale pre-training data. Therefore, JEPA4Rec is pre-trained using three key objectives: Masked Language Modeling (MLM), Mapping Representation, and Sequence-Item contrastive task.

#### 3.3.1 Masking strategy

User history sequences, represented as sentence sequences $\{S_1, S_2, \ldots, S_n\}$, are learned using the MLM approach, where text tokens are masked and predicted, but unlike traditional token-level learning, JEPA4Rec learns item embeddings in the representation space and extends MLM with a structured masking strategy.

Since we want the model to infer $i_{n+1}$ based on the user's history and partial information from $S_{n+1}$, we mask a significant portion (50%, detailed ablation study is in Appendix G) of tokens in $S_{n+1}$ and train the model to predict the full representation of $i_{n+1}$. For historical sentences, excessive masking could degrade the History Sequence Embedding $\mathbf{h}_{CLS}$, leading to information loss. Therefore, we apply a 15% token masking rate, consistent with pre-trained language model studies. For both history sequences and the next item, we apply the same masking strategy in BERT: (1) replacing tokens with [MASK] (80%), (2) replacing with a random token (10%), and (3) keeping the original token unchanged (10%). Following this principle, after masking tokens in both history item sentences and the next item, we utilize the history sequence embedding $\mathbf{h}_{CLS}$ to predict their full representations. However, since the number of masked tokens varies across user sequences, the number of masked items within each sequence also differs. To simplify this process, we sample one masked item per sequence and introduce a learnable zero vector when no tokens are masked.

After applying this masking strategy, we obtain the full representation of one masked item from the user's history (e.g., the $n$-th item in Figure 2) and the next item. These representations are then passed into the Target Encoder $f_{\bar\theta}$, where their embeddings are computed as:

$$\mathbf{h}_n = f_{\bar\theta}(\{[CLS], S_n\}), \quad \mathbf{h}_{n+1} = f_{\bar\theta}(\{[CLS], S_{n+1}\})$$

These encoded representations are then compared with their corresponding predictions generated by the Predictor to refine the learned item embeddings.

#### 3.3.2 Learning Framework

JEPA4Rec integrates the Mapping Representation task into the training process to enhance item representation learning. To achieve this, we propose a lightweight MLP-based Predictor, which utilizes the history sequence embedding $\mathbf{h}_{CLS}$ along with the Item Position Embedding $\mathbf{D}_n$, which corresponds to masked history items. This design enables the model to predict the full representation of $S_i$, mitigating information loss in $\mathbf{h}_{CLS}$ when encoding long sequences. The predicted representation is computed as:

$$\hat{\mathbf{h}}_n = \text{FFN}_1(\mathbf{h}_{CLS} \oplus \mathbf{D}_n)$$

Furthermore, since $\mathbf{h}_{CLS}$ must also infer the next item's representation based on partial information, we mask most tokens in $S_{n+1}$ to obtain $\mathbf{h}_{n+1,M}$. The combined information from $\mathbf{h}_{CLS}$ and $\mathbf{h}_{n+1,M}$ is then used to predict the full representation of $S_{n+1}$:

$$\hat{\mathbf{h}}_{n+1} = \text{FFN}_2(\mathbf{h}_{CLS} \oplus \mathbf{h}_{n+1,M})$$

Here, the concatenation operator ($\oplus$) is used to combine embeddings for simplicity. To help the model predict item representations accurately, we define the mapping loss and MLM loss as follows:

$$\mathcal{L}_{map} = \|\mathbf{h}_n - \hat{\mathbf{h}}_n\|^2 + \|\mathbf{h}_{n+1} - \hat{\mathbf{h}}_{n+1}\|^2, \quad \mathcal{L}_{MLM} = -\sum_{i=0}^{|\mathcal{V}|} y_i \log(p_i)$$

Where $p_i$ represents the predicted probability of the masked token belonging to the $i$-th vocabulary word, and $y_i$ is the corresponding ground truth label. The mapping loss enhances item representation learning, while the MLM loss bridges the semantic gap between the pretrained language model's knowledge and the textual information in the recommendation dataset.

Another pre-training objective for JEPA4Rec is the sequence-item contrastive task (S-I), commonly used for next-item prediction (Hou et al., 2022; 2023). We treat the next items in the ground truth as positive instances, while negative instances are selected using in-batch negatives instead of negative sampling or full softmax. In-batch negatives leverage ground-truth items from other sequences within the same batch, effectively serving as negative instances from multiple domains when training on large datasets. This approach not only reduces computational costs since JEPA4Rec generates item embeddings dynamically rather than maintaining an item embedding table, but also enhances model generalization across domains.

$$\mathcal{L}_{S-I} = -log \frac{e^{\text{sim}(\mathbf{h}_{CLS}, \mathbf{h}_{n+1})/\tau}}{\sum_{i \in B} e^{\text{sim}(\mathbf{h}_{CLS}, \mathbf{h}_i)/\tau}}$$

where sim is the cosine similarity score between 2 vectors; $\mathbf{h}_{n+1}$ is the representation of the ground truth next item; $\mathcal{B}$ is the ground truth item set in one batch; and $\tau$ is a temperature parameter. At the pre-training stage, we use a multi-task training strategy to jointly optimize JEPA4Rec:

$$\mathcal{L}_{PT} = \mathcal{L}_{S-I} + \lambda_1 \cdot \mathcal{L}_{MLM} + \lambda_2 \cdot \mathcal{L}_{map}$$

Where $\lambda_1, \lambda_2$ a hyperparameters to control the weight of the MLM and S-I task loss. The pre-trained model will be fine-tuned for new target domains.

### 3.4 Finetuning Framework

After pretraining, only the Context Encoder is used for fine-tuning in the target domain, with the learned user embedding encapsulating rich and transferable user preference information. We encode all item sentences to construct a dynamic, learnable item matrix $\mathcal{I}$, which enables probability computation for next-item prediction over the entire dataset:

$$P_{\mathcal{I}}(i_{n+1}|\mathbf{h}_{CLS}) = \text{Softmax}(\mathbf{h}_{CLS} \cdot \mathbf{h}_{n+1})$$

To reduce computational overhead, $\mathcal{I}$ is updated per epoch instead of every batch. For finetuning, we adopt the widely used cross-entropy loss and train the model with a sequence-item contrastive learning task using fully softmax over the entire dataset based on cosine similarity between items:

$$\mathcal{L}_{FT} = -\log \frac{e^{\text{sim}(\mathbf{h}_{CLS}, \mathbf{h}_{n+1})/\tau}}{\sum_{i \in I} e^{\text{sim}(\mathbf{h}_{CLS}, \mathbf{h}_i)/\tau}}$$

### 3.5 Discussion

We compare JEPA4Rec with prior sequential recommendation methods to highlight its key innovations and advantages. Traditional models such as SASRec (Kang & McAuley, 2018) and BERT4Rec (Sun et al., 2019) rely on trainable item ID embeddings, which are non-transferable across domains and struggle with cold-start problems due to data sparsity. Context-enhanced models like UniSRec (Hou et al., 2022), S3-Rec (Zhou et al., 2020), and VQ-Rec (Hou et al., 2023) attempt to address this by extracting item features from pretrained language models and injecting them into standalone sequential architectures. However,

these approaches typically separate textual understanding from sequential modeling and rely heavily on contrastive objectives or side information fusion.

JEPA4Rec addresses these limitations by leveraging JEPA to learn item-level representations directly in the embedding space, rather than at the token level. This enables the model to produce semantically rich and transferable embeddings that better capture user preferences and generalize across domains. Central to our approach is an information-rich user representation, analogous to the $[CLS]$ token in language models, learned through two auxiliary objectives: reconstructing historical item embeddings to preserve long-range preference signals, and recovering full item information from partially masked input to simulate real-world scenarios with incomplete metadata. As a result, JEPA4Rec is well-suited for settings with sparse or partial item descriptions, offering improved adaptability to new domains and cold-start items (see Appendix E for detailed experiments).

## 4 Experiments

### 4.1 Experimental Setup

**Datasets** To evaluate JEPA4Rec's performance, we conduct pre-training and fine-tuning using various Amazon review datasets (Hou et al., 2024). The dataset statistics after preprocessing are presented in Table 1. For pre-training, we utilize data from only three categories: *Automotive, Grocery and Gourmet Food, and Movies and TV*, accounting for approximately 35% of the dataset size used in prior studies (Li et al., 2023a; Hou et al., 2022; 2023). These categories serve as the source domain datasets.

| Datasets | #Users | #Items | #Inters. | Avg. n | Density |
|---|---|---|---|---|---|
| Pre-training | $115,778$ | $158,006$ | $1,250,489$ | $10.70$ | $6.8 \times 10^{-5}$ |
| Scientific | $11,041$ | $5,327$ | $76,896$ | $6.96$ | $1.3 \times 10^{-3}$ |
| Instruments | $27,530$ | $10,611$ | $231,312$ | $8.40$ | $7.9 \times 10^{-4}$ |
| Arts | $56,210$ | $22,855$ | $492,492$ | $8.76$ | $3.8 \times 10^{-4}$ |
| Office | $101,501$ | $27,932$ | $798,914$ | $7.87$ | $2.8 \times 10^{-4}$ |
| Pet | $47,569$ | $37,970$ | $420,662$ | $8.84$ | $2.3 \times 10^{-4}$ |
| *Online Retail* | $4,181$ | $3,896$ | $401,248$ | $9.75$ | $2.5 \times 10^{-2}$ |

Table 1: Dataset statistics

For finetuning, we test JEPA4Rec on five Amazon categories (Scientific, Instruments, Crafts, Office, Pet Supplies) to assess cross-domain generalization. Additionally, we use the Online Retail dataset[1], a UK e-commerce platform with no shared users, making it a more challenging cross-setting. Following Hou et al. (2022), we keep five-core datasets, filter out users/items with fewer than five interactions. Item text representations are built from title, categories, and brand (Amazon) or Description (Online Retail).

**Baselines** We compare JEPA4Rec against three categories of baseline models: (1) ID-only Methods: SASRec (Kang & McAuley, 2018), BERT4Rec (Sun et al., 2019); (2) ID-text Methods: S3-Rec (Zhou et al., 2020), LlamaRec; (3) Text-only Methods: UniSRec (Hou et al., 2022), VQ-Rec (Hou et al., 2023), RecFormer (Li et al., 2023a). Detailed descriptions of these models can be found in the Appendix C.

**Evaluation Settings** We use NDCG@10, Recall@10, and MRR as metrics, applying a leave-one-out strategy: the latest interaction for testing, the second latest for validation, and the rest for training. The ground-truth item is ranked among all items, and average scores are reported. To ensure a fair comparison with RecFormer, the state-of-the-art method, we adopt the same experimental settings as RecFormer, detailed in Appendix D. Other baselines follow prior work settings.

### 4.2 Overall Performance

Table 2 presents a comparative analysis of JEPA4Rec against baseline methods across six different datasets. Text-only methods consistently outperform ID-only and ID-text approaches on Amazon datasets. However, on the highly dense *Online Retail* dataset, ID-based methods remain effective. Notably, on Instruments, Arts, and Scientific datasets,

---

[1]https://www.kaggle.com/datasets/carrie1/ecommerce-data

| Scenario | Dataset | Metric | SASRec | BERT4Rec | LlamaRec | S³-Rec | UniSRec | VQ-Rec | RecFormer | JEPA4Rec | Improv |
|----------|---------|--------|--------|----------|----------|--------|---------|--------|-----------|----------|--------|
| Cross-Domain | Scientific | R@10 | 0.1305 | 0.1061 | 0.1180 | 0.0804 | 0.1255 | 0.1361 | 0.1684 | **0.1761** | 4.57% |
| | | N@10 | 0.0797 | 0.0790 | 0.0947 | 0.0451 | 0.0862 | 0.0843 | 0.1198 | **0.1282** | 7.01% |
| | | MRR | 0.0696 | 0.0759 | 0.0856 | 0.0392 | 0.0786 | 0.0712 | 0.1071 | **0.1190** | 11.11% |
| | Pet | R@10 | 0.0881 | 0.0765 | 0.1019 | 0.1039 | 0.0933 | 0.1002 | 0.1363 | **0.1471** | 7.92% |
| | | N@10 | 0.0569 | 0.0602 | 0.0781 | 0.0742 | 0.0702 | 0.0761 | 0.1086 | **0.1210** | 11.41% |
| | | MRR | 0.0507 | 0.0585 | 0.0719 | 0.0710 | 0.0650 | 0.0697 | 0.0940 | **0.1157** | 23.09% |
| | Instruments | R@10 | 0.0995 | 0.0972 | 0.1034 | 0.1110 | 0.1119 | 0.1289 | 0.1279 | **0.1347** | 4.49% |
| | | N@10 | 0.0634 | 0.0707 | 0.0767 | 0.0797 | 0.0785 | 0.0812 | 0.1001 | **0.1057** | 5.59% |
| | | MRR | 0.0577 | 0.0677 | 0.0689 | 0.0755 | 0.0740 | 0.0776 | 0.0958 | **0.1014** | 5.84% |
| | Arts | R@10 | 0.1342 | 0.1236 | 0.1337 | 0.1399 | 0.1333 | 0.1298 | 0.1797 | **0.1920** | 6.84% |
| | | N@10 | 0.0848 | 0.0942 | 0.0938 | 0.1026 | 0.0894 | 0.0912 | 0.1249 | **0.1442** | 15.45% |
| | | MRR | 0.0742 | 0.0899 | 0.0847 | 0.1057 | 0.0798 | 0.0878 | 0.1187 | **0.1341** | 12.97% |
| | Office | R@10 | 0.1196 | 0.1205 | 0.1201 | 0.1186 | 0.1262 | 0.1336 | 0.1559 | **0.1676** | 7.51% |
| | | N@10 | 0.0832 | 0.0972 | 0.0864 | 0.0911 | 0.0919 | 0.1011 | 0.1151 | **0.1276** | 10.86% |
| | | MRR | 0.0751 | 0.0932 | 0.0817 | 0.0957 | 0.0848 | 0.0912 | 0.1094 | **0.1185** | 8.31% |
| Cross-Platform | Online Retail | R@10 | 0.2275 | 0.1384 | 0.2361 | 0.2218 | 0.2284 | 0.2301 | 0.2355 | **0.2429** | 3.14% |
| | | N@10 | 0.0978 | 0.0478 | 0.1061 | 0.0954 | 0.0912 | 0.0913 | 0.1249 | **0.1266** | 1.36% |
| | | MRR | 0.0901 | 0.0332 | 0.0941 | 0.0858 | 0.0793 | 0.0865 | 0.0985 | 0.0985 | - |

Table 2: Performance comparison of recommendation methods across different datasets.

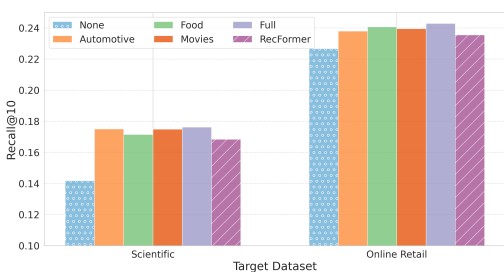

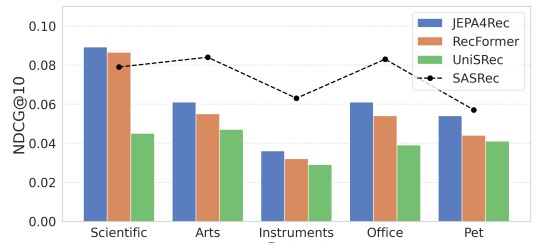

Figure 3: Performance (Recall@10) comparison w.r.t different pre-training datasets. *Full* denotes result pre-training with 3 datasets and *None* denotes the training from scratch.

Figure 4: Performance (NDCG@10) comparison of three text-only methods under the zero-shot setting.

text-based models achieve the highest performance, likely due to the rich descriptive metadata available for items.

JEPA4Rec outperforms all baseline models across datasets, except for the MRR metric on *Online Retail*. On average, it improves Recall@10 by 6.22% and NDCG@10 by 10.06%, demonstrating its effectiveness in recommendation tasks. The results highlight the advantages of JEPA4Rec's two-stage training strategy. The pre-training phase facilitates the learning of generalizable item representations, allowing the model to grasp common user preferences. The finetuning phase further enhances adaptability to new domains, as all items are represented through textual descriptions, enabling seamless transfer across different recommendation scenarios.

### 4.3 Efficient Learning Representation Performance

**Universal Pre-training** Figure 3 highlights the efficiency of JEPA4Rec's pre-training strategy. This figure demonstrates that JEPA4Rec pre-trained on three datasets outperforms models pre-trained on a single dataset and finetuned on the *Scientific* and *Online Retail* domains. Additionally, it surpasses the finetuned public checkpoint of RecFormer, which was pre-trained on seven Amazon datasets. Pre-training on multiple datasets allows the model to initialize with well-learned weights, leading to improved adaptation during finetuning. Notably, JEPA4Rec achieves strong results despite utilizing only 35% of the pre-training data compared to state of the art model RecFormer, demonstrating the robustness and efficiency of its pre-training approach.

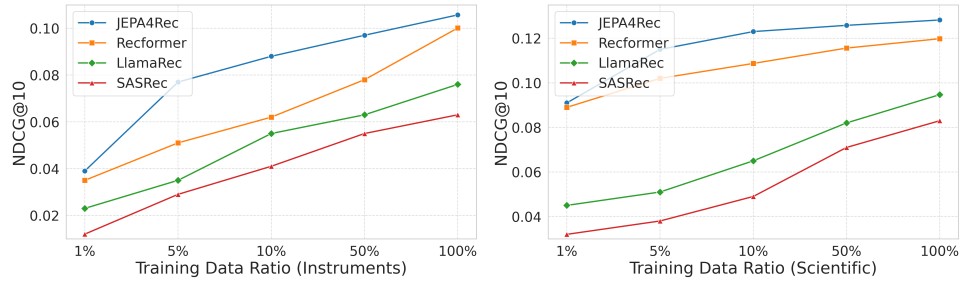

Figure 5: NDCG@10 comparison between models over different sizes of training data

| Variants | Scientific | | | Online Retail | | |
|---|---|---|---|---|---|---|
| | NDCG@10 | Recall@10 | MRR | NDCG@10 | Recall@10 | MRR |
| (0) JEPA4Rec | **0.1282** | **0.1761** | **0.1190** | **0.1266** | **0.2429** | **0.0985** |
| (1) w/o MLM loss | 0.1170 | 0.1653 | 0.1128 | 0.1118 | 0.2216 | 0.0858 |
| (2) w/o pre-training | 0.0935 | 0.1365 | 0.0855 | 0.1198 | 0.2185 | 0.0965 |
| (3) w/o token type emb. | 0.1251 | 0.1749 | 0.1072 | 0.1231 | 0.1755 | 0.0915 |

Table 3: Performance comparison across model variants

**Zero-shot** To ensure fair zero-shot evaluation, we re-trained RecFormer on the same three datasets as JEPA4Rec while keeping all hyperparameters identical. Since SASRec is ID-only and unsuitable for zero-shot settings, we trained it fully supervised on each target domain for comparison. Figure 4 demonstrates that JEPA4Rec outperforms other text-only models by $1 - 5\%$ across all five datasets. This shows that JEPA4Rec learns common-sense user preferences better than RecFormer and UniSRec. Notably, on Scientific, JEPA4Rec surpasses SASRec's fully supervised performance in zero-shot settings, highlighting the power of language-based recommendation.

**Low-Resource Training** We compare ID-based models (SASRec, LlamaRec) with text-only models (RecFormer, JEPA4Rec) across varying training data ratios on both the Instruments and Scientific datasets. As shown in Figure 5, text-only models consistently outperform ID-based ones, particularly under low-resource conditions (e.g., 1% or 5% of the data). This performance gap is attributed to the ability of language-based models to utilize semantic item descriptions during pre-training, enabling better generalization to unseen items. In contrast, ID-based models assign random embeddings to new items, making it difficult to generate meaningful recommendations with sparse data. As the training data increases, ID-based models such as SASRec and LlamaRec show noticeable improvements, reducing the performance gap. Nevertheless, JEPA4Rec remains the top-performing model across all data scales, highlighting the robustness of contrastive language modeling for recommendation.

## 4.4 Ablation Study

### 4.4.1 Study of JEPA4Rec

We perform an ablation study on *Scientific* (cross-domain) and *Online Retail* (cross-platform) datasets to assess JEPA4Rec's key components, with detailed results in Table 3. (1) w/o MLM loss: Removing Masked Language Modeling (MLM) reduces performance across all metrics, highlighting its role in bridging the semantic gap between Longformer's pre-trained knowledge and the recommendation dataset; (2) w/o pre-training: Performance drops significantly, emphasizing the necessity of pre-training for generalizable item embeddings and common-sense user preference learning; (3) w/o token type embeddings: While it has little impact on *Scientific*, it significantly lowers Recall@10 on *Online Retail*, showing its importance in distinguishing patterns within item sentences.

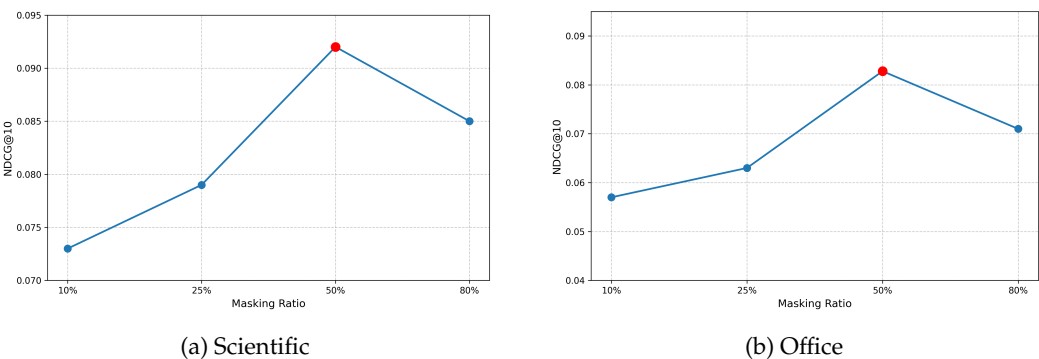

(a) Scientific            (b) Office

Figure 6: Effect of Masking Ratio on Performance

### 4.4.2 Masking Ratio Analysis

To investigate the effect of varying masking ratios on next-item prediction performance, we conducted a comprehensive ablation study by masking different proportions of tokens in next-item sentences during pre-training. Specifically, we trained our JEPA4Rec model from scratch on both the Scientific dataset (representing small-scale data) and the Office dataset (representing large-scale data). The results, averaged over five runs, are presented in Figure 6. We observe a consistent pattern across both datasets: the NDCG@10 score improves as the masking ratio increases from 10% to 50%, and then degrades when further increased to 80%. This trend suggests that moderate masking (around 50%) provides the most effective learning signal for the model, balancing information retention with learning generalizable representations.

### 4.4.3 Additional Experimental Results.

Further experiments are included in Appendix G, covering case studies, robustness analysis, and additional ablation. First, we examine JEPA4Rec's zero-shot capability and adaptability to partial item information. Case studies show that JEPA4Rec can accurately rank relevant items with minimal text cues, significantly outperforming RecFormer. We also provide extended zero-shot evaluations across six datasets, where JEPA4Rec remains competitive despite being pre-trained on fewer domains. Lastly, ablation studies on token embedding components and loss functions demonstrate the importance of MLM loss, token structure embeddings, and contrastive learning, each contributing significantly to model effectiveness.

## 5 Conclusion

In this work, we propose JEPA4Rec, a novel framework that integrates Joint Embedding Predictive Architecture (JEPA) with language modeling to enhance sequential recommendation. By representing items as text sentences and leveraging a bidirectional Transformer encoder, JEPA4Rec learns semantically rich and transferable item representations while improving common-sense user preference modeling. Our framework incorporates a novel masking strategy and a two-stage training approach to enhance recommendation accuracy and adaptability across domains. Extensive experiments on six real-world datasets demonstrate that JEPA4Rec outperforms state-of-the-art methods, particularly in cross-domain, cross-platform, and low-resource scenarios, while requiring significantly less pre-training data. These results highlight the effectiveness and efficiency of our approach in learning generalizable and robust item representations for sequential recommendation.

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

## A  Joint Embedding Predictive Architecture

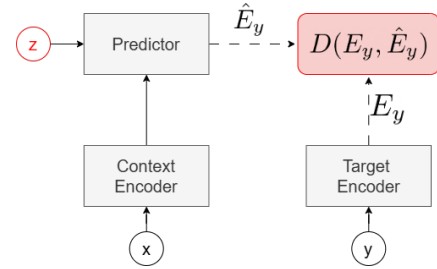

JEPA's key advantage lies in predicting abstract representations instead of raw pixel or token space, allowing the model to focus on meaningful semantic features rather than low-level details (LeCun, 2022). As shown in Figure 7, its architecture consists of an encoder $f_\theta(\cdot)$ to compute input representations and a predictor $g_\phi(\cdot)$ to estimate the representation of $y$ based on $x$ and an auxiliary variable $z$, which captures transformations between them. The model minimizes the discrepancy between predicted and actual embeddings via $D\left(E_y, \text{Pred}\left(E_x, z\right)\right)$, using an asymmetric **Context** and **Target** Encoder to prevent representation collapse (Chen & He, 2021). By operating in representation space rather than raw input, JEPA focuses on learning meaningful, generalizable features and capturing underlying data relationships for robust self-supervised learning.

Figure 7: Joint-Embedding Predictive Architectures aim to estimate the representation of a target input $y$ based on the representation of a context input $x$, leveraging a predictor network that incorporates auxiliary variable $z$ to enhance prediction performance.

## B  Details of the Tokenizer Component

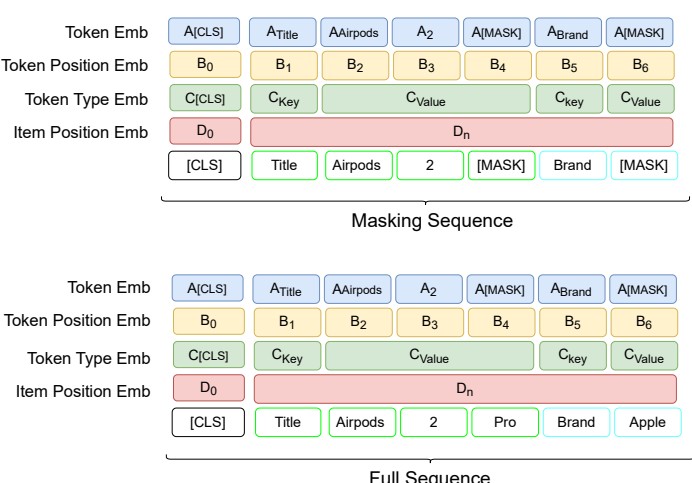

Figure 8: Tokenizer Component

Figure 8 illustrates the Tokenizer Component in detail. This component is responsible for transforming item metadata into sequence representations through a masking strategy that simulates partial observation. Specifically, it flattens item metadata and applies token-level masking to create a masked input sequence, which is then passed through the Context Encoder. Simultaneously, the complete (unmasked) input is processed by the Target Encoder. These parallel encoding paths enable the model to learn robust representations by contrasting partial and full information.

In detail, to encode item sentences, we follow previous work in Sun et al. (2019); Geng et al. (2022) to construct four different types of embeddings:

- Token Embedding represents the corresponding tokens, denoted by $\mathbf{A} \in \mathbb{R}^{V_w \times d}$, where $V_w$ is the number of words in our vocabulary and $d$ is the embedding dimension. JEPA4Rec represents items using text instead of ID tokens (Hua et al., 2023), making its size independent of the number of items and ensuring flexibility across different recommendation scenarios.

- Token position embedding represents the position of a token in a sequence, denoted by $\mathbf{B} \in \mathbb{R}^d$. It is designed to help Transformer-based models capture the sequential structure of words.

- Token type embedding identifies the origin of a token within the input. It consists of three vectors, $\mathbf{C}_{CLS}, \mathbf{C}_{\text{Attribute}}, \mathbf{C}_{\text{Value}} \in \mathbb{R}^d$, which distinguish whether a token belongs to $[CLS]$, attribute names, or values. In recommendation datasets where attribute keys are often repeated across items, token type embedding enables the model to recognize and differentiate these recurring patterns.

- Item Position Embedding represents the position of items in a sequence, with all tokens from sentences $S_i$ represented as $\mathbf{D}_i \in \mathbb{R}^d$ and the entire item position embedding matrix as $\mathbf{D} \in \mathbb{R}^{n \times d}$, where $n$ is the maximum length of a user's interaction sequence. $\mathbf{D}$ facilitates the alignment between word tokens and their corresponding items.

## C  Baselines

We compare JEPA4Rec against three categories of baseline models:

- ID-only Methods: SASRec (Kang & McAuley, 2018) utilizes a directional self-attention mechanism to capture item correlations within a sequence. BERT4Rec (Sun et al., 2019) applies a bidirectional Transformer with a cloze-style objective for modeling user behavior.

- ID-text Methods: S3-Rec (Zhou et al., 2020) leverages mutual information maximization for pre-training sequential models, capturing relationships between attributes, items, subsequences, and full sequences. LlamaRec is a two-stage ranking framework that leverages LLMs by retrieving candidate items via small-scale recommenders and ranking them using a verbalizer-based approach.

- Text-only Methods: UniSRec (Hou et al., 2022) employs text-based item representations from a pre-trained language model, adapting to new domains through an MoE-enhanced adaptor. VQ-Rec (Hou et al., 2023) mitigates the over-reliance on textual features in transferable recommenders by mapping item text to discrete codes, which are then used to retrieve item representations from a code embedding table (text → code → representation). We initialize them with pre-trained parameters provided by the authors and fine-tune them on target domains. RecFormer (Li et al., 2023a) introduces a framework that formulates items as text sequences and employs a bidirectional Transformer to learn language representations for sequential recommendation.

## D  Experiment Settings

We use a finetuning batch size of 16, a learning rate of 5e-5, token limits of 32 per attribute and 1024 per sequence, a maximum of 50 items per sequence, a temperature parameter $\tau = 0.05$, and an MLM loss weight $\lambda_1 = 0.1$. For pre-training, we use a batch size of 32 and Mapping Representation loss weight $\lambda_2 = 0.1$.

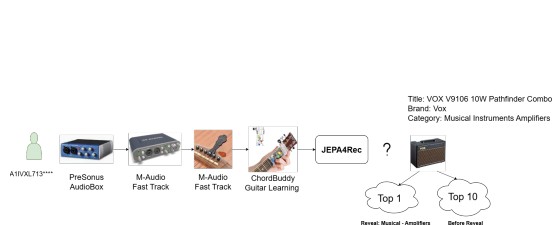
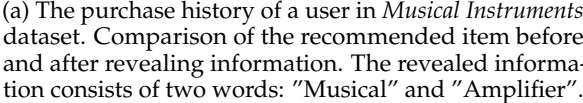
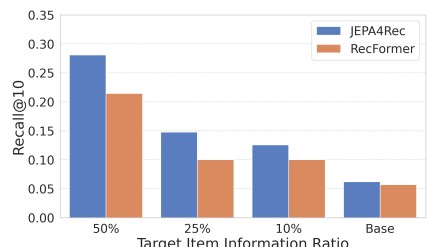

(a) The purchase history of a user in *Musical Instruments* dataset. Comparison of the recommended item before and after revealing information. The revealed information consists of two words: "Musical" and "Amplifier".

(b) Testing the zero-shot capability of JEPA4Rec when revealing different ratios of words for the target item sentences in *Musical Instruments*.

Figure 9: Case study and analysis of revealing information

# E   Case Study

**Firt case study**, we analyze how text-only models adapt with partial next-item information by revealing different word ratios from item sentences in a zero-shot setting on the Instruments dataset, detailed in Figure 9. Both JEPA4Rec and RecFormer improve Recall@10, but JEPA4Rec shows a larger gain due to its mapping loss pre-training. Notably, JEPA4Rec's performance doubles with just 10% of item information, demonstrating its ability to leverage minimal textual cues. For a user interested in music production and guitar-related items, JEPA4Rec ranked the ground-truth item 10th initially but moved it to 1st after revealing partial information. This highlights its strong adaptability, making it well-suited for real-world scenarios where only limited item context is available.

**Second case study**, to evaluate the robustness of our approach when item information is incomplete, we conducted a systematic evaluation by randomly dropping 20%, 40%, and 60% of tokens from item sequences at the target domain during the fine-tuning process. This simulates real-world scenarios where item descriptions may be incomplete or partially available. The masking is applied to the flattened item sentences containing "Title ... Brand ... Category ..." information.

| Masking Ratio | Method | Scientific (NDCG@10) | Instrument (NDCG@10) | Office (NDCG@10) |
|---|---|---|---|---|
| Full info | RecFormer | 0.1198 | 0.1001 | 0.1151 |
| | JEPA4Rec | **0.1282** | **0.1057** | **0.1276** |
| 20% masked | RecFormer | 0.1145 | 0.0971 | 0.1207 |
| | JEPA4Rec | **0.1223** | **0.0999** | **0.1242** |
| 40% masked | RecFormer | 0.1114 | 0.0954 | 0.1174 |
| | JEPA4Rec | **0.1204** | **0.0997** | **0.1233** |
| 60% masked | RecFormer | 0.1088 | 0.0905 | 0.1093 |
| | JEPA4Rec | **0.1201** | **0.0979** | **0.1197** |

Table 4: Robustness evaluation with different masking ratios

These results demonstrate JEPA4Rec's superior robustness when making recommendations based on partial item information. Notably, JEPA4Rec maintains stable performance even when 60% of item tokens are masked, with only modest performance degradation (e.g., from 0.1282 to 0.1201 on the Scientific dataset). In contrast, RecFormer shows a significant accuracy downgrade as the masking ratio increases, with performance declining more substantially (e.g., from 0.1198 to 0.1088 on the Scientific dataset). This robustness stems from JEPA4Rec's ability to learn item representations in the embedding space through its mapping loss pre-training, enabling effective inference even with incomplete textual descriptions. JEPA4Rec consistently outperforms RecFormer by $\sim 3-5\%$ on the NDCG metric across all masking scenarios.

# F  More zero-shot Experiments

| Datasets | | JEPA4Rec | RecFormer |
|---|---|---|---|
| | NDCG@10 | 0.0896 | **0.0897** |
| Scientific | Recall@10 | **0.1319** | 0.1310 |
| | MRR | 0.0810 | **0.0820** |
| | NDCG@10 | 0.0360 | **0.0416** |
| Instruments | Recall@10 | 0.0632 | **0.0699** |
| | MRR | 0.0316 | **0.0359** |
| | NDCG@10 | 0.0610 | **0.0722** |
| Arts | Recall@10 | 0.1061 | **0.1090** |
| | MRR | 0.0592 | **0.0620** |
| | NDCG@10 | **0.0568** | 0.0524 |
| Pet | Recall@10 | **0.0782** | 0.0698 |
| | MRR | **0.0525** | 0.0473 |
| | NDCG@10 | **0.0610** | 0.0551 |
| Office | Recall@10 | **0.0840** | 0.0823 |
| | MRR | 0.0462 | **0.0472** |
| | NDCG@10 | 0.0310 | 0.0310 |
| Online Retail | Recall@10 | 0.0490 | **0.0592** |
| | MRR | **0.0318** | 0.0310 |

Table 5: Performance comparison between JEPA4Rec and RecFormer under zero-shot setting

We conducted a comparison of JEPA4Rec's zero-shot capability when pre-trained on three Amazon datasets against the official public checkpoint of RecFormer, which was pre-trained on seven Amazon datasets. We can observe that the recommendation capabilities of the two models are quite similar, demonstrating JEPA4Rec's robust training ability that does not depend on large-scale data.

# G  More ablation studies

## G.1  Token Embedding Component Analysis

We evaluate the importance of different token embedding types:

**Experiment 1: Removing MLM Loss** We remove the MLM loss component to assess how much semantic information learning from textual data contributes to JEPA4Rec's recommendation performance.

**Experiment 2: Removing Token Type Embedding ($\mathcal{C}$)** This experiment evaluates the importance of token type embedding in distinguishing different types of textual information within item representations.

**Experiment 3: Removing Token Position Embedding ($\mathcal{B}$) and MLM Loss** Token position embedding $\mathcal{B}$ is crucial for language models to determine token positions. We remove both $\mathcal{B}$ and MLM loss to assess JEPA4Rec's recommendation capability when losing components that help learn semantic information from e-commerce text data.

**Experiment 4: Removing MLM Loss, Token Position Embedding ($\mathcal{B}$), and Token Type Embedding ($\mathcal{C}$)** This experiment determines JEPA4Rec's performance when relying solely on item representation recovery through L2 Mapping loss.

**Note on Item Position Embedding ($\mathcal{D}$):** We do not remove $\mathcal{D}$ as it is essential for determining item positions in history sequences and enabling JEPA4Rec to recover complete item representations.

These results demonstrate:

1. **MLM loss is crucial for performance** as it helps the model learn additional semantic information from the textual data of items. The performance drop when removing

| Variant | Scientific | | | Online Retail | | |
|---|---|---|---|---|---|---|
| | NDCG@10 | Recall@10 | MRR | NDCG@10 | Recall@10 | MRR |
| Full JEPA4Rec | **0.1282** | **0.1761** | **0.1190** | **0.1266** | **0.2429** | **0.0985** |
| w/o MLM loss | 0.1170 | 0.1653 | 0.1128 | 0.1118 | 0.2216 | 0.0858 |
| w/o Token Type ($\mathcal{C}$) | 0.1251 | 0.1749 | 0.1072 | 0.1231 | 0.1755 | 0.0915 |
| w/o MLM + Token Pos ($\mathcal{B}$) | 0.1102 | 0.1542 | 0.1021 | 0.1094 | 0.1898 | 0.0824 |
| w/o MLM + $\mathcal{B}$ + Token Type ($\mathcal{C}$) | 0.0967 | 0.1398 | 0.0889 | 0.1019 | 0.1687 | 0.0792 |

Table 6: Token embedding component analysis

MLM loss (from 0.1282 to 0.1170 NDCG@10 on the Scientific dataset) shows its significant contribution to understanding item semantics.

2. **All embedding components are important** for the model's effectiveness:

- **Token type embedding ($\mathcal{C}$)** provides discriminative power for different attribute patterns, with noticeable performance degradation when removed.
- **Token position embedding ($\mathcal{B}$)** is essential for understanding token sequence structure.
- The combined removal of multiple components leads to progressively worse performance, confirming that each embedding type contributes unique and valuable information to the model.

### G.2 Ablation Studies on Loss Components

We conduct ablation studies on different loss components to understand their contributions:

**Experiment 1: Pre-training when dropping contrastive loss** We evaluate JEPA4Rec's ability to learn item representations in the embedding space by using only the Mapping Loss and MLM during pre-training. This experiment demonstrates how well the item representation recovery mechanism alone contributes to recommendation performance without the contrastive learning component.

**Experiment 2: Fine-tuning with BPR Loss** During fine-tuning, we replace the contrastive loss with the well-known Bayesian Personalized Ranking (BPR) loss to assess the impact of different ranking objectives:

$$\mathcal{L}_{BPR} = -\sum_{(u,i,j) \in D_S} \ln \sigma(\hat{r}_{ui} - \hat{r}_{uj})$$

where $D_S = \{(u,i,j)|i \in I_u^+ \land j \in I \setminus I_u^+\}$ represents the training data with positive item $i$ and one sampled negative item $j$ for user $u$, and $\hat{r}_{ui}$ denotes the predicted preference score.

| Variant | Scientific | | | Online Retail | | |
|---|---|---|---|---|---|---|
| | NDCG@10 | Recall@10 | MRR | NDCG@10 | Recall@10 | MRR |
| Full JEPA4Rec | **0.1282** | **0.1761** | **0.1190** | **0.1266** | **0.2429** | **0.0985** |
| Pre-training: Drop contrastive loss | 0.1216 | 0.1634 | 0.1087 | 0.1064 | 0.1853 | 0.0892 |
| Fine-tuning: BPR Loss | 0.1245 | 0.1723 | 0.1156 | 0.0934 | 0.1637 | 0.0865 |

Table 7: Ablation studies on loss components

These results demonstrate that **contrastive loss with its ability to sample multiple negative items remains well-suited for JEPA4Rec**. The performance degradation when replacing contrastive loss with BPR loss (which samples only one negative item per positive) shows that the multi-negative sampling strategy of contrastive learning provides richer training signals. This is particularly evident in the Online Retail dataset, where BPR loss shows significant performance drops across all metrics, highlighting the importance of diverse negative sampling for effective representation learning in our framework.

