# OpenReview forum: "Learning Effective Language Representations for Sequential Recommendation via Joint Embedding Predictive Architecture"
_colmweb.org/COLM/2025/Conference — COLM 2025_

### Official Review · Reviewer_fZrx · 2025-04-26

**Rating:** 7
**Confidence:** 4
**Ethics Flag:** 1

**Summary:**

In this paper, the authors propose JEPA4Rec, a framework that combines Joint Embedding Predictive Architecture with language modeling of item textual descriptions. Specifically, it represents items as text sentences by flattening descriptive information, and encodes these sentences through a bidirectional Transformer encoder with modified embedding layers tailored for capturing item information in recommendation datasets. Through extensive experiments, the authors demonstrate the benefits of the proposed model over a set of existing baselines.

**Reasons To Accept:**

The proposed model is easy to follow and technically sound. The authors have conducted extensive experiments to demonstrate the benefits of the proposed model over a set of existing baselines across various experiment settings.

**Reasons To Reject:**

While I do not see any major flaws in this paper, I feel that there are a number of places that the authors can further improve this paper: (1) it seems that flattering the metadata attributes is an important component in the proposed framework, so will the order of these attributes affect the model outcomes? (2) I wonder if we can rely on the "knowledge" within the LLMs to encode the items directly for those cases where the metadata attributes are not available (3) I wonder if the authors can conduct additional analysis to study the relative contribution of four types of embeddings in the proposed model; additionally, why do the authors choose to simply sum them together? It seems that a more intuitive way is to do a weighted sum.

---

> ### Author Response · Authors · 2025-06-02
> **Rebuttal by Authors**
>
> We sincerely thank the reviewer for the positive and constructive feedback. We appreciate your thoughtful suggestions for improving our work. Below, we present additional ablation studies and analyses to address your concerns in detail.
> > (Q1) it seems that flattering the metadata attributes is an important component in the proposed framework, so will the order of these attributes affect the model outcomes?
>
> **Answer:** We conducted experiments by changing the order of item sequences during encoding from "Title ... Brand ... Category" to "Category ... Brand ... Title". We then performed pre-training and fine-tuning on Scientific and Office datasets with the same experimental setup as in our paper. The results are as follows:
>
> | Attribute Order | Scientific (NDCG@10) | Office (NDCG@10) |
> |-----------------|---------------------|------------------|
> | Title → Brand → Category | 0.1282 | 0.1276 |
> | Category → Brand → Title | 0.1276 | 0.1278 |
>
> The results show minimal impact (~1-2% difference), indicating that JEPA4Rec is robust to attribute ordering due to its bidirectional Transformer architecture.
>
> > (Q2) I wonder if we can rely on the "knowledge" within the LLMs to encode the items directly for those cases where the metadata attributes are not available
>
> **Answer:** We conducted experiments using only the Title of items as textual data for encoding, then performed pre-training and fine-tuning on Scientific and Office datasets. The results are:
>
> | Input Information | Scientific (NDCG@10) | Office (NDCG@10) |
> |-------------------|---------------------|------------------|
> | Full metadata (Title + Brand + Category) | 0.1282 | 0.1276 |
> | Title only | 0.1209 | 0.1239 |
>
> While using only titles results in some performance degradation, JEPA4Rec still maintains reasonable performance. Additionally, please refer to Section 2 "Robustness to Partial Information" where we tested JEPA4Rec's recommendation capability when masking textual data, demonstrating its ability to work with incomplete information.
>
> >(Q3) I wonder if the authors can conduct additional analysis to study the relative contribution of four types of embeddings in the proposed model; additionally, why do the authors choose to simply sum them together? It seems that a more intuitive way is to do a weighted sum.
>
> **Answer:** We have provided ablation studies on the importance of different token embeddings in Section 3 of our Global Rebuttal. We use simple summation similar to BERT's token embedding approach. To clarify how weighted summation affects JEPA4Rec's performance, we conducted experiments with weighted embedding:
>
> $$\mathbf{E}_w = \text{LayerNorm}(\mathbf{A}_w + \mathbf{B}_w + \mathbf{C}_w + 1.25 \times \mathbf{D}_w)$$
>
> We assign higher weight to Item Position Embedding ($\mathbf{D}_w$) considering the important role of it in JEPA4Rec's architecture. After pre-training and fine-tuning on Scientific and Online Retail datasets with the same experimental settings, the results are:
>
> | Embedding Strategy | Scientific (NDCG@10) | Online Retail (NDCG@10) |
> |--------------------|---------------------|-------------------------|
> | Simple sum (original) | 0.1282 | 0.1266 |
> | Weighted sum (1.25×D) | 0.1284 | 0.1273 |
>
> The weighted approach shows slight improvement, suggesting that emphasizing different type of embedding information can be beneficial, though the simple summation approach already performs well.
>
> We appreciate your feedback and are happy to clarify further if needed.

---

> > ### Comment · Reviewer_fZrx · 2025-06-02
> >
> > Thank you for the additional clarification. I have increased my score accordingly.

---

> > > ### Author Response · Authors · 2025-06-03
> > > **Thank you**
> > >
> > > Thanks for your thoughtful feedback and for taking the time to reconsider your score. We truly appreciate your constructive comments, which helped us clarify key aspects of our work and improve the overall presentation.

---

### Official Review · Reviewer_HQmU · 2025-05-23

**Rating:** 7
**Confidence:** 3
**Ethics Flag:** 1

**Summary:**

The paper is well structured and well written, and its main contribution is the adoption of the JEPA framework in the sequential recommendation framework, named JEPA4Rec. JEPA4Rec exploits textual description of items to capture the common sense knowledge related to user preferences based on their consumption histories.

Originality: This paper is the first at using the JEPA framework in the NLP and recommendation field, and I can some see novelty in this aspect, in addition to the customization proposed by the authors that make JEPA4Rec different from the original framework. However, this could still be considered a limited novel work, as the paper needs more details about the challenges that previous works did not tackle, while this paper did.

Quality and clarity: the paper is written in a clear way and is quite easy to read, although the constraints imposed by the page limit are evident in the discussion of some parts, in particular, the related works section is too short.
For example, in the introduction section, the authors claim that "the semantic gap between modalities remains a challenge". This is not true, as there are some works in the field of recommendation that show how modalities alignment tend to improve the recommendation performance, such as LMM4Rec [1].

Significance: The reasons and practical impact of the proposed methodology are not clear and should be more emphasized: which is the problem the paper is solving, and why previous work did not solve it? The authors should have been more clear in this aspect and better frame their work in the literature, although the experiments and the results confirm their approach.

Other comments:

In the pre-training phase, during the masking strategy, the authors claim to mask "a significant portion (50%)" of the tokens of the item to be recommended; it would be interesting to consider this portion as a sensitivity parameter, since it is not indicated why they choose this particular number and not, for example, 80% - that can still be considered as a significant portion. Or, if the authors based this choice on other works, they should declare that.

Another aspect I'm not convinced is related to the performance evaluation.
Here, a common metric used in sequential recommendation models is the hit rate @k (hr@k), with k typically set to k={1,5} (for example, the mentioned models Bert4Rec S3-Rec user hr@1 among the other metrics).
Since the focus of sequential recommendation is next-item prediction, hr@1 is necessary, as it tells how good the model is at recommending exaclty the next item. This make sense also based on what the authors say in the Evaluation Settings paragraph (Section 4.1), as they applied a "leave-one-out strategy: the latest interaction for testing, the second latest for validation, and the rest for training". On the other hand, ndcg@10 and recall@10 might be not enough good at capturing the behavior of the model in the exact next item prediction task, but "only" consider what happens within the next 10 recommended items.

[1]: Zixuan Yi, Zijun Long, Iadh Ounis, Craig Macdonald, and Richard McCreadie. 2025. Enhancing Recommender Systems: Deep Modality Alignment with Large Multi-Modal Encoders. ACM Trans. Recomm. Syst. 3, 4, Article 52 (December 2025), 25 pages. https://doi.org/10.1145/3718099

Edit: after discussion, the authors made some aspects of the paper more clear and added other metrics, and I increased my review score.

**Reasons To Accept:**

- Paper well written and clear
- Source code and dataset processing script provided

**Reasons To Reject:**

- It can be considered a paper with limited novelty
- It needs more detailed context and discussion about the practical impact
- Evaluation setting and metric not necessarily suitable for the task

---

> ### Author Response · Authors · 2025-06-02
> **Rebuttal by Authors**
>
> Thank you for the insightful and balanced review. We value the reviewer’s comments and respond to the key concerns below.
> > However, this could still be considered a limited novel work, as the paper needs more details about the challenges that previous works did not tackle, while this paper did.
>
> **Answer**: We have presented the novelty of JEPA4Rec in our Global Rebuttal. We summarize the key points as follows:
>
> 1. **Better pre-training framework** based on item embedding recovery mechanism
> 2. **Learning at item representation level** instead of only token level
> 3. **Experimental validation** demonstrating JEPA4Rec's effectiveness through comprehensive experiments
>
> The key challenges we address that previous works did not tackle include: (1) learning generalizable item representations across domains through embedding space recovery, (2) capturing common-sense user preferences through joint embedding predictive architecture, and (3) achieving effective cross-domain transfer with minimal pre-training data.
>
> > The related works section is too short.
>
> **Answer**: We will expand the related work section in future versions to provide more comprehensive coverage of relevant literature.
>
> > The claim about "semantic gap between modalities remains a challenge" contradicts works like LMM4Rec that show modality alignment improvements.
>
> **Answer**: Our statement is based on several previous works [1,2] that utilized different modalities for recommendation systems but did not achieve significant effectiveness improvements. We thank you for the evidence and will cite LMM4Rec in our related work section to provide a more balanced perspective on modality alignment progress.
>
> > Why choose 50% masking for the next item instead of other percentages like 80%
>
> **Answer**: We chose 50% masking for ground truth item tokens based on ablation studies presented in Appendix F (Page 14). Our experiments showed that 50% provides the optimal balance between providing sufficient challenge for the model while maintaining enough information for meaningful learning.
>
> >Missing Hit Rate@K metrics, particularly HR@1 for next-item prediction evaluation.
>
> **Answer**: We thank you for this suggestion. First, we clarify the metric definitions:
>
> - **Hit Rate@K**: Calculates the share of users for which at least one relevant item is present in the top K
> - **Recall@K**: Measures the coverage of relevant items in the top K
> - **NDCG@K**: Considers both relevance and position of items in the ranked list
>
> For the leave-one-out evaluation strategy, since we predict only 1 ground truth item, Recall equals Hit Rate in this case. We provide additional Hit Rate@1 results:
>
> | Method |Scientific (HR@1)|Online Retail (HR@1)| Office (HR@1) |
> |--------|---------------------|------------------|---------------|
> | RecFormer | 0.0845 | 0.0279 | 0.0243 |
> | JEPA4Rec | 0.0879 | 0.0305 | 0.0272 |
>
> JEPA4Rec outperforms RecFormer across all datasets, demonstrating the effectiveness of JEPA4Rec in next item recommendation tasks.
>
> > Need more detailed context about practical impact.
>
> **Answer**: JEPA4Rec can be widely applied in e-commerce, especially for platforms with multiple product domains like Amazon. Instead of requiring separate recommendation systems for each domain, we can train JEPA4Rec with moderate datasets from various domains and leverage its knowledge generalization capability for cross-domain recommendations without extensive retraining.
>
> **Key practical benefits:**
>
> 1. **Cost reduction**: Serving one JEPA4Rec model for multiple domains instead of domain-specific models
> 2. **Enhanced search experience**: Superior recommendation performance when revealing product information that customers are searching for (e.g., keywords about title, brand, or category)
> 3. **Robustness to incomplete data**: Effective recommendations even when item information is missing or incomplete
>
> Please see our Appendix D Case Study and Section 2 of the Global Rebuttal for detailed applications of JEPA4Rec with incomplete item data.
>
> We appreciate your feedback and are happy to clarify further if needed.
>
> [1] Yuan, Zheng, et al. "Where to go next for recommender systems? id-vs. modality-based recommender models revisited." Proceedings of the 46th International ACM SIGIR Conference on Research and Development in Information Retrieval. 2023.
>
> [2] Liu, Junling, et al. "Is chatgpt a good recommender? a preliminary study." arXiv preprint arXiv:2304.10149 (2023).

---

> > ### Author Response · Authors · 2025-06-03
> > **Call for Your Response**
> >
> > Dear Reviewer HQmU,
> >
> > Thank you once again for your valuable and constructive feedback on our paper.
> >
> > As we approach the end of the Rebuttal Phase (ending June 10 at 11:59 PM AoE), we would like to check if you have any final questions or concerns that we can address before the deadline. We are happy to clarify or expand on any aspect of the submission to assist your review process.
> >
> > Best regards,
> >
> > Authors

---

> > > ### Author Response · Authors · 2025-06-05
> > > **Another Call for Your Response**
> > >
> > > Dear Reviewer HQmU,
> > >
> > > Thank you for your thoughtful and detailed feedback on our paper.
> > >
> > > As we approach the end of the Discussion Phase (ending June 10 at 11:59 pm AoE), we want to check if you have any final questions. We would be glad to address them before the deadline.
> > >
> > > If you agree that our responses have adequately addressed the concerns you raised, we kindly ask you to consider whether increasing your score would more accurately reflect your revised evaluation of our paper. Thank you once again for your time and insightful comments!
> > >
> > > Best regards,
> > >
> > > Authors

---

> > > > ### Comment · Reviewer_HQmU · 2025-06-07
> > > > **Rebuttal received**
> > > >
> > > > Dear authors,
> > > >
> > > > I have read your rebuttal; thank you for the additional details you provided, and apologize for the delay in the answer (the emails were marked as spam, unfortunately).
> > > >
> > > > Results on Recall/Hit@1 are really interesting, and I suggest you to include them in the final version of the paper too (if you already have some results of Hit@k with other values for k, please add them as well).
> > > >
> > > > Similarly, I invite you add a short sentence that empirically justifies the choice related to the 50% of the masking tokens.
> > > >
> > > > The same holds for the expanded discussion on the practical impact.
> > > >
> > > > Given the authors answer, that clarified some aspects of the paper, I will increase my rating.

---

> > > > > ### Author Response · Authors · 2025-06-08
> > > > > **Thank you very much**
> > > > >
> > > > > Thank you very much for your feedback and for increasing your rating based on our clarifications. We truly appreciate your constructive suggestions regarding the inclusion of Recall/Hit@1 results and additional details on masking and practical impact. We will carefully incorporate these points and any additional results (such as Hit@k with k=1) in the final version of the paper to further strengthen its clarity and completeness.

---

### Official Review · Reviewer_F2XD · 2025-05-25

**Rating:** 7
**Confidence:** 4
**Ethics Flag:** 1

**Summary:**

This paper proposes a framework that uses textual item descriptions for the task of sequential recommendation. It explores representing items by their attributes and learning to encode them, to capture general item embeddings and subsequently exploit them to improve model performance on item recommendation in large-scale shopping datasets.

The paper provides a large set of experiments showing a range of results, detailed explanations for the pre-training framework and item embeddings. Notably, the experiments show that their trained model have a higher data efficiency than previous works, outperforming them in metrics at similar data ratios. However, the paper suffers from a lack of clarity overall, and the metric improvements are very small, questioning some of the claims of the paper and its originality.

UPDATE AFTER REBUTTAL: increased the score from 5 to 7 based on the authors' response, see my full response below.

**Questions To Authors:**

1. Can you expand your motives or support for the contributions?
2. How is a two-stage training approach a contribution and how does it differ from standard NLP in this paper?
3. How does the encoding item mechanism differ from previous works?
4. Can you provide further information on the difference in training/fine-tuning from standard NLP practices? Are you applying the MLM loss for pre-training and then not for fine-tuning?
5. When are the Predictor and Target trained? Are these frozen during fine-tuning?

**Reasons To Accept:**

- The paper provides in-depth details of the framework, easing its reproducibility.
- It shows iterative improvements over previous state-of-the-art methods, across domains and at different ratios of training data.
- The idea of reconstructing item information from partial descriptions described in the paper seems to allow learning generalisable item representations while exploiting text biases from model pre-training, potentially opening further avenues of research, particularly with larger models than explored in the paper.
- Overall, the paper is solid and may be technologically impactful, releasing resources that would benefit future works in the community.

**Reasons To Reject:**

- The main reason for rejecting this is the lack of clarity at many stages of the paper, particularly in Section 3. It is not obvious how each piece fits together, and requires re-reading paragraphs several times to fully understand what it is being explained. Further arguments for this point:
    - Figure 2 does not help, as it is too high-level to explain what is being described, yet does not clearly identify the parts of the system (i.e., how do the Predictor and the Target encoder fit together in the final model? What exactly is being used to predict the final output?).
    - This section mixes what is a contribution (or novel) and what is from previous works in the framework, leaving it up to the reader to extract the novelty (i.e., line 97 suggests that their masked language modelling is novel).
    - The section fluctuates between explaining low-level and high-level concepts without clear motives. For example, concepts that are perhaps obvious in NLP are described in detail or mentioned, without a particular goal or any modifications to their standard use, such as token embeddings (line 124), masked language modelling, or how BERT and RoBERTa can also be used for encoding (lines 154). I think the paper would benefit from moving these less-relevant parts to the Appendix to focus on the actual contribution.
- Some of the paper’s claims are too broad and not justified by the experiments. For example, in many parts of the paper, it claims that this framework learns common-sense user preferences (line 95, 163, 277, 296). However, I think that the framework aids a model to learn more general item representations, and thus is able to provide better recommendation predictions. I am not sure that the model learns common-sense (in the way described in the paper) or user preferences in general, and none of the experiments show this, particularly since the metric improvements against previous works are so small (~0.01 better).
- Discussion is very shallow and does not distil (or interpret) the results. The ablation experiments are also limited, despite all the different parts introduced in Section 3. For example, only token type embeddings are ablated, despite introducing 4 types in Section 3.2. I would expect that a paper providing a framework with so many different parts would provide stronger support for these design choices, such as ablating the sequence-item contrastive task, or the mapping loss (and not just the MLM loss).
- Novel contributions and originality are not clear throughout the paper, and the contributions themselves seem to lack support or justification. The main contributions are the encoder with modified embedding layers, the masking strategy and a 2-stage (training/fine-tuning), but neither of these is novel, and the differences with standard practices in NLP are very subtle and not explained in detail. I think the paper would benefit from rephrasing or re-structuring some parts to emphasise the contributions, perhaps targeted Figures in these sections.
- Very limited related works section and discussion of how their framework compares to previous works (§3.5).

---

> ### Author Response · Authors · 2025-06-02
> **Rebuttal by Authors**
>
> We thank the reviewer for the detailed and constructive feedback. We appreciate the thoughtful assessment of our paper's strengths as well as the concerns raised. Below we address the key points raised, particularly around the clarity of Section 3 and the novelty of our contributions.
> > It is not obvious how each piece fits together, and requires re-reading paragraphs several times to fully understand what it is being explained.
>
> **Answer**: We provide a comprehensive overview of the JEPA4Rec framework flow:
>
> **Framework Components:** JEPA4Rec consists of 3 main components:
> - **Context Encoder** ($f_\theta$): Processes masked item sequences
> - **Target Encoder** ($f_{\overline{\theta}}$): Processes complete item information (same architecture as Context Encoder, updated via Exponential Moving Average of Context Encoder parameters, not trained directly)
> - **Predictor** ($g_\phi$): Recovers full representations from partial information
>
> **Processing Flow:**
>
> 1. **Input Processing:** Item metadata is flattened $\rightarrow$ masking strategy creates partial information
>  $$\text{Masked Input} \rightarrow f_\theta \rightarrow h_{CLS}, h_{n+1,M}$$
> $$\text{Complete Input} \rightarrow f_{\overline{\theta}} \rightarrow h_n, h_{n+1}$$
> 2. **Representation Recovery:** Predictor recovers full representations:
>
> $\hat{h_n} = g_\phi(h_{CLS} \oplus \mathcal{D}_n)$
>
> $\hat{h_{n+1}} = g_\phi(h_{CLS} \oplus \mathcal{D}_{n+1})$
>
> Please note that we use light weight MLP for building $g_\phi$.
>
> 3. **Training:** The mapping loss encourages the predicted item embedding $\hat{h_n}$, which is derived from partial item textual data, to approximate the target embedding $h_n$, which is obtained from the full item textual information. Similarly, it aligns $\hat{h_{n+1}}$ with $h_{n+1}$ in the same manner. We also use MLM loss to help the model learn the semantic information of the dataset, and contrastive loss for the recommendation task.
>
> 4. **Fine-tuning:** Only Context Encoder is used for target domain recommendation with learned $h_{CLS}$ containing rich user preference information.
>
> > The section fluctuates between explaining low-level and high-level concepts without clear motives.
>
> **Answer**:  We agree. We will streamline common NLP terminology and move detailed explanations to the Appendix in future versions to focus on our actual contributions.
>
> > Claims about learning "common-sense user preferences" are too broad and not justified by experiments.
>
> **Answer**:  Thank you for pointing this out. The term "common-sense" follows the definition from Yann LeCun's work [1]. You are correct that we learn better item representations that create more informative $h_{CLS}$ embeddings (representing user preferences), leading to improved experimental results. Based on surveys of prominent sequential recommendation papers [2,3,4], an accuracy improvement of ~0.01 and average 5-10% improvement per dataset compared to SOTA RecFormer represents substantial progress in the recommendation system field.
>
> > Discussion is shallow and ablation experiments are limited despite introducing many components.
>
> **Answer**:  We have supplemented additional ablation studies in our Global Rebuttal, including ablation of sequence-item contrastive task and JEPA4Rec's components. We will provide more detailed discussion for each experiment in future versions.
>
> > Novel contributions and originality are not clear, and the differences with standard NLP practices are subtle.
>
> **Answer**: We appreciate the opportunity to clarify this. We have presented JEPA4Rec's novelty in our Global Rebuttal:
> 1. **Better pre-training framework** based on item embedding recovery mechanism
> 2. **Learning at item representation level** instead of only token level
> 3. **Experimental validation** demonstrating effectiveness through comprehensive experiments
>
> The key novelty lies in applying Joint Embedding Predictive Architecture to textual sequential recommendation, which has never been done before.
>
> > Very limited related works section and discussion of framework comparison.
>
> **Answer**:  We agree and will expand this section in future versions.
>
> We will continue to respond to the remaining questions in the next response due to length constraint.
>
> [1] LeCun, Yann. "A path towards autonomous machine intelligence version 0.9. 2, 2022-06-27." Open Review 62.1 (2022): 1-62.
>
> [2] Kang, Wang-Cheng, and Julian McAuley. "Self-attentive sequential recommendation." 2018 IEEE international conference on data mining (ICDM). IEEE, 2018.
>
> [3] Sun, Fei, et al. "BERT4Rec: Sequential recommendation with bidirectional encoder representations from transformer." Proceedings of the 28th ACM international conference on information and knowledge management. 2019.
>
> [4] Yue, Zhenrui, et al. "Linear recurrent units for sequential recommendation." Proceedings of the 17th ACM international conference on web search and data mining. 2024.

---

> > ### Author Response · Authors · 2025-06-02
> > **Rebuttal by Authors Part 2**
> >
> > > Q1: Can you expand your motives or support for the contributions?
> >
> > **Answer**: Thank you for the question. Our contributions are motivated by three key limitations in existing work:
> > 1. **Cross-domain transfer limitations** with ID-based representations.
> > 2. **Insufficient utilization of textual information** in e-commerce recommendation.
> > 3. **Lack of common-sense user preference learning** in current frameworks.
> >
> > JEPA4Rec addresses these through: (1) text-based item representation enabling cross-domain transfer, (2) JEPA architecture for better embedding space learning. We hope this explanation helps clarify the rationale behind our proposed contributions.
> >
> > > Q2: How is a two-stage training approach a contribution and how does it differ from standard NLP?
> >
> > **Answer**: Thank you for raising this point. The key difference is our addition of the **L2 Mapping Loss** and the use of a Joint Embedding Predictive Architecture. This design allows the pre-training process to create generalizable item embeddings and supports user preference learning—unlike standard NLP pre-training which typically focuses only on language modeling or understanding. We believe this structural difference is crucial for adapting to recommendation tasks.
> >
> > > Q3: How does the encoding item mechanism differ from previous works?
> >
> > **Answer**: We appreciate your observation. While the encoding components may appear similar to those in prior work, our ablation studies show that these components are essential within our overall framework. The novelty lies in their integration with the JEPA architecture and the use of mapping loss to recover semantically meaningful representations, which we believe is a key differentiator.
> >
> > > Q4: Can you provide further information on the difference in training/fine-tuning from standard NLP practices?
> >
> > **Answer**: Certainly, and thank you for the opportunity to elaborate.
> >
> > **Pre-training differences:**
> > - Standard NLP-based sequential recommendation: MLM loss + task-specific loss.
> > - JEPA4Rec: MLM loss + contrastive loss + **L2 Mapping Loss** (a novel addition).
> >
> > **Architecture differences:**
> > - JEPA4Rec trains the Predictor alongside the Context Encoder.
> > - The Target Encoder is updated via Exponential Moving Average (EMA), rather than being directly trained.
> > - The MLM loss is only applied during pre-training to capture semantic information from item textual data.
> >
> > **Fine-tuning phase:**
> > - Only the Context Encoder is fine-tuned using contrastive loss for downstream recommendation tasks.
> > - The Predictor and Target Encoder are not used during fine-tuning.
> >
> > We hope these distinctions clarify the uniqueness of our training pipeline compared to standard NLP procedures.
> >
> > > Q5: When are the Predictor and Target trained? Are these frozen during fine-tuning?
> >
> > **Answer**: Thank you for this technical question. The training flow is as follows:
> >
> > - **Predictor**: Trained jointly with the Context Encoder during the pre-training phase.
> > - **Target Encoder**: Not directly trained. Instead, it is updated as an Exponential Moving Average (EMA) of the Context Encoder's parameters.
> > - **Fine-tuning**: Only the Context Encoder is used and fine-tuned on the recommendation task; the Predictor and Target Encoder are not involved in this phase.
> >
> > We appreciate your thoughtful questions and are happy to clarify further if needed.

---

> > > ### Author Response · Authors · 2025-06-03
> > > **Call for Your Response**
> > >
> > > Dear Reviewer F2XD,
> > >
> > > Thank you once again for your valuable and constructive feedback on our paper.
> > >
> > > As we approach the end of the Rebuttal Phase (ending June 10 at 11:59 PM AoE), we would like to check if you have any final questions or concerns that we can address before the deadline. We are happy to clarify or expand on any aspect of the submission to assist your review process.
> > >
> > > Best regards,
> > >
> > > Authors

---

> > > > ### Author Response · Authors · 2025-06-05
> > > > **Another Call for Your Response**
> > > >
> > > > Dear Reviewer F2XD,
> > > >
> > > > Thank you for your thoughtful and detailed feedback on our paper.
> > > >
> > > > As we approach the end of the Discussion Phase (ending June 10 at 11:59 pm AoE), we want to check if you have any final questions. We would be glad to address them before the deadline.
> > > >
> > > > If you agree that our responses have adequately addressed the concerns you raised, we kindly ask you to consider whether increasing your score would more accurately reflect your revised evaluation of our paper. Thank you once again for your time and insightful comments!
> > > >
> > > > Best regards,
> > > >
> > > > Authors

---

> ### Comment · Reviewer_F2XD · 2025-06-06
> **Rebuttal Response**
>
> Based on the authors' rebuttal and comments, I have increased my score from 5 to 7.
>
> I think that the overall paper is good and, although the novelty is sometimes difficult to distinguish from previous works, the authors have done an excellent job at addressing the questions and providing additional information (particularly ablations). The novelty may not be on the individual components, but their combination and thus the framework are the novel part. The experiments are solid and have gone beyond the average paper.
>
> I have some concerns that the additional information provided by the authors will not easily fit into the final version of the paper. I think that the paper requires large edits to some sections to incorporate the clarifications in the global rebuttal, as the framework and some components explained here differ significantly from the paper (i.e., the processing flow is not clear in the paper).

---

> > ### Author Response · Authors · 2025-06-06
> > **Thank you very much**
> >
> > Thank you very much for your thoughtful feedback and for increasing your score based on our rebuttal and clarifications. We sincerely appreciate your constructive comments and your recognition of the strengths of our framework and experiments. Your suggestions regarding improving the clarity and flow of the paper are very valuable, and we will carefully revise the final version to better reflect the key components and processing flow discussed in the rebuttal.

---

### Author Response · Authors · 2025-06-02
**Global Rebuttal**

Dear AC and reviewers,

Thank you for your constructive feedback on our submission. We appreciate the time and effort you have dedicated to evaluating our work. Below, we address your main concerns and provide additional analyses and experimental results.

---
## 1. Novelty and Technical Contributions
Traditional NLP-based recommendation models typically rely on Masked Language Modeling (MLM) to learn token-level semantics from item descriptions and optimize a single loss function to predict the next item. While effective to some extent, these models often fail to learn high-level item representations that generalize well across domains or adapt efficiently when only partial item information is available. Moreover, token-level learning may not fully capture the rich preference signals embedded in user interaction histories.

**JEPA4Rec addresses these limitations by introducing a novel approach based on the Joint Embedding Predictive Architecture (JEPA), which directly learns item-level representations in the embedding space rather than at the token level.** This shift allows JEPA4Rec to produce generalizable and semantically rich item embeddings that are more robust to domain shifts and more expressive in modeling user preferences. Our goal is to construct an information-rich user representation (analogous to the [CLS] token in language models) that captures common sense of user preferences. To this end, JEPA4Rec introduces two key auxiliary learning objectives beyond the standard contrastive loss used in sequential recommendation:
- Reconstructing embeddings of historical (past) items, helping to preserve and re-integrate preference signals that may be lost when encoding long interaction sequences.
- Recovering complete item information from partial input, where only parts of a future (ground truth) item’s textual description are visible.

This simulates realistic recommendation settings where item metadata is incomplete or revealed incrementally, and encourages the model to make accurate predictions even under limited information. To support these goals, JEPA4Rec applies a structured masking strategy:
- For history sequences, 15% of tokens are masked to encourage contextual learning without excessive information loss.
- For the next-item prediction, a 50% masking ratio is used to make the task sufficiently challenging, forcing the model to rely on both the user sequence and partial item data.

A lightweight MLP-based predictor is then used to fuse the user representation, item position encoding, and the masked ground truth item. The prediction is matched against the true item embedding via a regression loss, with targets encoded by a frozen encoder whose parameters are updated using Exponential Moving Average (EMA) from the context encoder, following techniques inspired by self-supervised learning in computer vision. Therefore, JEPA4Rec is not only suitable for standard sequential recommendation scenarios but also particularly effective in more complex recommendation settings like:
- Only partial item information is available (e.g., title without full metadata), requiring the model to quickly adapt to new items.
- The textual data is sparse or incomplete, where traditional token-level MLM approaches often fall short.

These advantages are further illustrated and validated through additional experiments presented in Section 2 below.
## 2. Robustness to Partial Information
**JEPA4Rec demonstrates superior recommendation capability with partial information** due to its embedding recovery pre-training process. We provide evidence through our Case Study (Appendix D, page 13) showing performance when revealing partial ground truth item information.

**Additional experiment:** To evaluate the robustness of our approach when item information is incomplete, we conducted systematic evaluation by randomly dropping 20%, 40%, and 60% of tokens from item sequences at the target domain during fine-tuning process. This simulates real-world scenarios where item descriptions may be incomplete or partially available. The masking is applied to the flattened item sentences containing "Title ... Brand ... Category ..." information.

**Table 1**:  Performance under masked item information settings
| Masking Ratio | Method | Scientific (NDCG@10) | Instrument (NDCG@10) | Office (NDCG@10) |
|---------------|--------|---------------------|------------------|------------------|
| Full info     | RecFormer | 0.1198 | 0.1001 | 0.1151 |
|               | JEPA4Rec | **0.1282**|**0.1057** | **0.1276** |
| 20% masked    | RecFormer | 0.1145 | 0.0971 | 0.1207 |
|               | JEPA4Rec | **0.1223** |**0.0999** | **0.1242** |
| 40% masked    | RecFormer | 0.1114 | 0.0954 | 0.1174 |
|               | JEPA4Rec | **0.1204** | **0.0997** | **0.1233** |
| 60% masked    | RecFormer | 0.1088 | 0.0905 | 0.1093 |
|               | JEPA4Rec |**0.1201** | **0.0979** | **0.1197** |

Due to length constraints, we will continue in the next response.

---

> ### Author Response · Authors · 2025-06-02
> **Global Rebuttal Part 2**
>
> We now proceed to discuss the results presented in Table 1. These results demonstrate JEPA4Rec's superior robustness when making recommendations based on partial item information. Notably, JEPA4Rec maintains stable performance even when 60% of item tokens are masked, with only modest performance degradation (e.g., from 0.1282 to 0.1201 on Scientific dataset). In contrast, RecFormer shows significant accuracy downgrade as the masking ratio increases, with performance declining more substantially (e.g., from 0.1198 to 0.1088 on Scientific dataset). This robustness stems from JEPA4Rec's ability to learn item representations in the embedding space through its mapping loss pre-training, enabling effective inference even with incomplete textual descriptions. JEPA4Rec consistently outperforms RecFormer by ~3-5% on NDCG metric across all masking scenarios.
> ## 3. Extended Ablation Studies
> We acknowledge the current ablation study section is concise due to page limitations. We provide comprehensive additional ablation studies below:
> ### 3.1 Token Embedding Component Analysis
> We evaluate the importance of different token embedding types:
> - **Experiment 1: Removing MLM Loss** We remove the MLM loss component to assess how much semantic information learning from textual data contributes to JEPA4Rec's recommendation performance.
> - **Experiment 2: Removing Token Type Embedding ($\mathcal{C}$)** This experiment evaluates the importance of token type embedding in distinguishing different types of textual information within item representations.
> - **Experiment 3: Removing Token Position Embedding ($\mathcal{B}$) and MLM Loss** Token position embedding $\mathcal{B}$ is crucial for language models to determine token positions. We remove both $\mathcal{B}$ and MLM loss to assess JEPA4Rec's recommendation capability when losing components that help learn semantic information from e-commerce text data.
> - **Experiment 4: Removing MLM Loss, Token Position Embedding ($\mathcal{B}$), and Token Type Embedding ($\mathcal{C}$)** This experiment determines JEPA4Rec's performance when relying solely on item representation recovery through L2 Mapping loss.
>
> **Note on Item Position Embedding ($\mathcal{D}$):** We do not remove $\mathcal{D}$ as it is essential for determining item positions in history sequences and enabling JEPA4Rec to recover complete item representations.
>
> **Table 2**: Ablation study on the importance of embedding components
> | Variant | Scientific |  |  | Online Retail |  |  |
> |---------|-----------|--|--|---------------|--|--|
> |         | **NDCG@10** | **Recall@10** | **MRR** | **NDCG@10** | **Recall@10** | **MRR** |
> | Full JEPA4Rec | **0.1282** | **0.1761** | **0.1190** | **0.1266**| **0.2429** | **0.0985** |
> | w/o MLM loss | 0.1170 | 0.1653 | 0.1128  | 0.1118 | 0.2216 | 0.0858 |
> | w/o Token Type ($\mathcal{C}$) | 0.1251 | 0.1749 | 0.1072 | 0.1231| 0.1755 | 0.0915|
> | w/o MLM + Token Pos ($\mathcal{B}$) | 0.1102 | 0.1542 | 0.1021 | 0.1094 | 0.1898 | 0.0824 |
> | w/o MLM + $\mathcal{B}$ + Token Type ($\mathcal{C}$) | 0.0967 | 0.1398 | 0.0889 | 0.1019 | 0.1687 | 0.0792 |
> These results demonstrate:
> 1. **MLM loss is crucial for performance** as it helps the model learn additional semantic information from textual data of items. The performance drop when removing MLM loss (from 0.1282 to 0.1170 NDCG@10 on Scientific dataset) shows its significant contribution to understanding item semantics.
> 2. **All embedding components are important** for the model's effectiveness:
>    - **Token type embedding ($\mathcal{C}$)** provides discriminative power for different attribute patterns, with noticeable performance degradation when removed.
>    - **Token position embedding ($\mathcal{B}$)** is essential for understanding token sequence structure.
>    - The combined removal of multiple components leads to progressively worse performance, confirming that each embedding type contributes unique and valuable information to the model.

---

> > ### Author Response · Authors · 2025-06-02
> > **Global Rebuttal Part 3**
> >
> > ### 3.2 Ablation Studies on Loss Components
> > We conduct ablation studies on different loss components to understand their contributions:
> > - **Experiment 1: Pre-training when dropping contrastive loss** We evaluate JEPA4Rec's ability to learn item representations in the embedding space by using only the Mapping Loss and MLM during pre-training. This experiment demonstrates how well the item representation recovery mechanism alone contributes to recommendation performance without the contrastive learning component.
> > - **Experiment 2: Fine-tuning with BPR Loss** During fine-tuning, we replace the contrastive loss with the well-known Bayesian Personalized Ranking (BPR) loss to assess the impact of different ranking objectives:
> > $$L_{BPR} = -\sum_{(u,i,j)\in D_S}\text{ln}\sigma(r_{ui}-r_{uj})$$
> > where $D_S=\{(u, i, j) | i \in I_u^+ \land j \in I \setminus I_u^+\}$ represents the training data with positive item $i$ and one sampled negative item $j$ for user $u$, and $r_{ui}$ denotes the predicted preference score.
> >
> > **Table 3**: Ablation study on recommendation loss
> > | Variant | Scientific |  |  | Online Retail |  |  |
> > |---------|-----------|--|--|---------------|--|--|
> > |         | **NDCG@10** | **Recall@10** | **MRR** | **NDCG@10** | **Recall@10** | **MRR** |
> > | Full JEPA4Rec | **0.1282** | **0.1761** | **0.1190** | **0.1266** | **0.2429** | **0.0985** |
> > | Pre-training: Drop contrastive loss | 0.1216 | 0.1634 | 0.1087 | 0.1064 | 0.1853 | 0.0892 |
> > | Fine-tuning: BPR Loss | 0.1245 | 0.1723 | 0.1156 | 0.0934 | 0.1637 | 0.0865 |
> >
> > These results demonstrate that **contrastive loss with its ability to sample multiple negative items remains well-suited for JEPA4Rec**. The performance degradation when replacing contrastive loss with BPR loss (which samples only one negative item per positive) shows that the multi-negative sampling strategy of contrastive learning provides richer training signals. This is particularly evident in the Online Retail dataset, where BPR loss shows significant performance drops across all metrics, highlighting the importance of diverse negative sampling for effective representation learning in our framework.
> >
> > ## 4. Related Work and Discussion
> > Due to page limitations, the Related Work section and Section 3.5 Discussion in our current version are concise. We will expand these sections in future versions. Here, we summarize the key differences between JEPA4Rec and previous works:
> > ### 4.1 Advantages of Text-based Item Representation
> > JEPA4Rec leverages textual descriptions of items to learn generalizable representations across multiple domains and enables easy knowledge transfer to new domains. This approach is more advanced than using only discrete item IDs [1, 2], which limits knowledge transfer between domains and fails to utilize the rich textual data available in E-commerce platforms.
> > ### 4.2 Novel Application of JEPA to Sequential Recommendation
> > JEPA4Rec is the first work to apply Joint Embedding Predictive Architecture with textual data for sequential recommendation, with the goal of learning common-sense user preferences through generalizable item representations. The key difference lies in our pre-training process: instead of using only MLM loss for semantic information learning and contrastive loss for recommendation tasks [3, 4], JEPA4Rec introduces L2 Mapping Loss that enables the model to learn in the embedding space by using encoded user history vectors to recover item information, thereby helping the model learn user preferences more effectively. This unique combination allows JEPA4Rec to:
> > - Learn generalizable item representations beyond token-level understanding.
> > - Recover complete item information from partial textual cues.
> > - Achieve superior cross-domain generalization with minimal pre-training data.
> >
> > [1] Kang, Wang-Cheng, and Julian McAuley. "Self-attentive sequential recommendation." 2018 IEEE international conference on data mining (ICDM). IEEE, 2018.
> >
> > [2] Sun, Fei, et al. "BERT4Rec: Sequential recommendation with bidirectional encoder representations from transformer." Proceedings of the 28th ACM international conference on information and knowledge management. 2019.
> >
> > [3] Li, Jiacheng, et al. "Text is all you need: Learning language representations for sequential recommendation." Proceedings of the 29th ACM SIGKDD Conference on Knowledge Discovery and Data Mining. 2023.
> >
> > [4] Hou, Yupeng, et al. "Learning vector-quantized item
> > representation for transferable sequential recommenders." Proceedings of the ACM Web Conference 2023. 2023.
> >
> > ---
> >
> > We hope this rebuttal addresses the raised concerns and further clarifies the novelty and strength of our approach. We are happy to provide additional analyses or clarifications upon request.

---

### Author Response · Authors · 2025-06-08
**Summary of Planned Revisions**

Based on the valuable feedback from the reviewers and the additional analyses we have conducted, we summarize here the main improvements and clarifications we plan to incorporate in the upcoming version of the paper:
- We will expand the discussion of the novelty and technical contributions of JEPA4Rec, emphasizing its embedding-space learning paradigm (to be updated in Section 3 and Related Work).
- We will include the new experimental results on robustness to partial item information, which demonstrate JEPA4Rec’s strong performance under varying levels of missing item information (to be added in an appendix and summarized in the main text).
- We will incorporate a more comprehensive set of ablation studies on the contributions of different token embeddings and loss components, providing a clearer understanding of their roles (to be detailed in an appendix).
- We will add an explicit empirical justification for the choice of 50% masking ratio in the next-item prediction objective.
- We will revise and expand the Related Work section to more clearly position JEPA4Rec relative to prior methods in sequential and text-based recommendation.
- Additional clarifications and improvements in presentation will be made across the paper to address the reviewers' comments.

We thank the reviewers again for their constructive feedback, which will help us substantially improve the clarity and quality of the upcoming version.

---

### Decision · Program_Chairs · 2025-07-08

**Decision:**

Accept

**Comment:**

JEPA4Rec is a new method for sequential recommendation that uses a Joint Embedding Predictive Architecture (JEPA) and language modeling of item descriptions to overcome data sparsity and better understand user preferences. It works by treating item information (like title and category) as text, encoding it with a Transformer, and then using masked prediction and a two-stage self-supervised training to learn rich, transferable item representations. This approach significantly improves recommendation performance, especially in situations with limited data or across different domains.

Reviewer F2XD highlights how this paper would benefit from some major edits that would help it clarify the overall contribution of the architecture and what the main changes are to the standard Jepa architecture. Some of these have been addressed by the authors’ response to this reviewer, and I would suggest that the authors make sure that the final version of the paper takes all of this into account.

Reviewer HQmU highlighted an important lack of justification of certain parameters, such as the masking ratio, as well as details regarding the evaluation metrics used. The authors provided a detailed response and added additional results which clarify the contribution. I would invite the authors to take all these comments into account and refine the final version accordingly.

Reviewer fZrx and F2XD highlight important weaknesses regarding ablations in the experimental evaluation. For instance, the role played by the different types of embeddings or the different attribute order is not clearly studied. However, during the rebuttal phase, the authors have promptly reported additional experiments that seem to justify the choices they have made. I would invite the authors to report this in the paper (potentially in the appendix) to make sure that this weakness is mitigated.

Overall, all the reviewers agree that this represents an interesting methodological contribution for the NLP and RecSys communities. There are many changes that have been highlighted by the reviewers, which the authors have promised to take into account in the final version of the paper.